# Daily steps offset risks of sedentary behavior in the All of Us research program

Neil S. Zheng[1], Shi Huang[2], Jeffrey Annis[3,4,5], Hiral Master[3,4,6], Kelsie M. Full[7] & Evan L. Brittain [3,4,8] ✉

Sedentary behavior is associated with increased mortality and chronic diseases, yet it remains unclear whether higher daily step counts can mitigate these risks. In this study, we analyzed longitudinal sedentary and step data from Fitbit devices in the All of Us Research Program to examine incident diagnoses of chronic conditions. We show that greater sedentary time was associated with higher risk of obesity, diabetes mellitus, hypertension, coronary artery disease, heart failure, chronic kidney disease, metabolic dysfunction-associated steatotic liver disease, chronic obstructive pulmonary disease, major depressive disorder, sleep apnea, and atrial fibrillation. Increasing daily steps offset the excess risk of high sedentary time (14 vs. 8 hours/day) for several conditions, with the additional steps required ranging from 1700 to 5500 per day. However, no step count fully offset sedentary risks for coronary artery disease or heart failure. These findings support personalized, behavior-based recommendations that consider both sedentary behavior and daily steps.

Prolonged sedentary behavior is associated with all-cause mortality and chronic diseases, particularly cardiometabolic diseases[1–6]. Sedentary behavior is defined as any waking behavior with low energy expenditure (≤1.5 metabolic equivalents), such as sitting or reclining[4]. Globally, over one-third of adults have insufficient physical activity and prolonged daily sedentary behavior[7,8]. Replacing sedentary time with physical activity may reduce the risk of cardiometabolic disease and mortality[2,9].

The relationship between sedentary behavior, daily steps, and chronic disease remains understudied. Most prior research has focused on moderate-to-vigorous physical activity (MVPA) in relation to mitigating chronic disease risk and prolonging optimal health[2,9]. U.S. guidelines for physical activity recommend MVPA (≥150 min per week) but offer no quantitative guidance on sedentary time or daily steps[10]. The increasingly widespread use of commercial wearable

devices (greater than 1 in 4 Americans) and smartphones that can track step count allows patients to monitor step count as a more accessible measure of physical activity[11]. Several studies have suggested that daily steps and MVPA are highly correlated and may have similar associations with mortality and cardiovascular disease[12–16]. These studies have typically recommended between 7000 and 9000 daily steps for the general population to reduce risk of chronic disease and mortality[2,12–15]. However, it remains unclear whether individuals with greater sedentary behavior need additional steps to offset sedentary-associated risks for chronic diseases. Recently, Ahmadi et al. reported that roughly 9000 to 10,500 steps per day was associated with the lowest risk of incident cardiovascular disease or mortality independent of sedentary behavior using actigraphy data collected over 2 weeks in the UK Biobank[13]. However, shorter duration actigraphy-based measures of daily count and sedentary

[1]Brigham and Women's Hospital, Boston, MA, USA. [2]Department of Biostatistics, Vanderbilt University Medical Center, Nashville, TN, USA. [3]Vanderbilt Institute for Clinical and Translational Research, Vanderbilt University Medical Center, Nashville, TN, USA. [4]Department of Medicine, Center for Digital Genomic Medicine, Vanderbilt University Medical Center, Nashville, TN, USA. [5]Department of Medicine, Division of Genetic Medicine, Vanderbilt Genetics Institute, Vanderbilt University Medical Center, Nashville, TN, USA. [6]Division of Physical Therapy, School of Medicine, University of North Carolina at Chapel Hill, Chapel Hill, NC, USA. [7]Department of Medicine, Division of Epidemiology, Vanderbilt University Medical Center, Nashville, TN, USA. [8]Division of Cardiovascular Medicine, Vanderbilt University Medical Center, Nashville, TN, USA. ✉e-mail: evan.brittain@vumc.org

behavior cannot capture longitudinal, natural behavior due to seasonal or individual variations as well as potential observer bias. Additionally, prior studies have focused on mortality and cardiovascular disease, and there have been limited reports on the joint effect of daily steps and sedentary behavior on other common chronic diseases, such as obesity, type 2 diabetes mellitus, and hypertension[13].

The All of Us research program aims to gather health data from nearly one million persons living in the United States (U.S.)[17]. Participants share multiple longitudinal sources of health-related information, including electronic health records (EHRs), physical measures, and data from Fitbit devices, which are commercial wearable activity trackers that provide objective longitudinal measurements of sedentary time and daily steps from participants. Linking Fitbit data with EHRs enables large-scale epidemiological studies of objective wearable metrics and clinical outcomes[15,18,19]. Understanding how longitudinal patterns in sedentary behavior and daily steps relate to chronic diseases may support more intuitive and accessible physical activity recommendations[20,21].

In this work, we leverage the All of Us dataset and time-varying analyzes to examine associations between longitudinal sedentary time and the incidence of chronic diseases. We also estimate the potential for daily step count to offset the health risks associated with prolonged sedentary behavior.

## Results

Among the 58,527 individuals with Fitbit activity data, 15,327 adult participants with linked EHR data and sleep data met inclusion criteria for analysis, resulting in a total of 13,682,755 days of observation (Supplementary Fig. 1). The median age was 51.7 years (interquartile range [IQR]: 37.3–63.5). Most participants were female (72.0%) and White (79.7%). The median duration of Fitbit monitoring was 3.7 years (IQR: 1.7–6.5). Median daily sedentary time was 11.6 h (IQR: 10.6–12.5) and median daily step count was 7416 (IQR: 5506–9663) (Table 1). Significant differences in median daily sedentary time and steps were observed across demographic and lifestyle factors, including age, sex, self-reported race, education, and alcohol use. Patients who were older, identified as Black, or did not have a college degree had higher daily sedentary time and lower daily step counts.

We conducted time-varying Cox proportional hazard analyzes to evaluate the association between sedentary time and 12 selected chronic diseases. Higher sedentary time (75th vs 25th percentile) was associated with increased risk for 11 of 12 conditions, including obesity (hazard ratio: 1.45; 95% confidence interval: 1.37–1.53), diabetes mellitus (1.55; 1.36–1.78), hypertension (1.21; 1.16–1.27), atrial fibrillation (1.25; 1.10–1.43), heart failure (1.46; 1.26–1.70), CAD (1.18; 1.05–1.32), CKD (1.23; 1.04–1.44), MASLD (1.40; 1.24–1.58), COPD (1.66; 1.16–2.38), MDD (1.15; 1.07–1.23), and sleep apnea (1.41; 1.30–1.51). The association between sedentary time and ischemic stroke did not reach statistical

## Table 1 | Baseline characteristics of study participants

| Variable | Study Participants [a] | Sedentary time (Hours) [b] | P-value [c] | Steps (thousands) [b] | P-value |
|---|---|---|---|---|---|
| Age | | | <0.001 | | <0.001 |
| 18–39 | 4556 (29.7%) | 11.4 (10.5–12.3) | | 7.6 (5.9–9.5) | |
| 40–59 | 5728 (37.4%) | 11.6 (10.6–12.5) | | 7.5 (5.6–9.9) | |
| 60–79 | 4926 (32.1%) | 11.8 (10.7–12.8) | | 7.2 (4.1–9.7) | |
| ≥80 | 117 (0.8%) | 12.5 (11.5–13.3) | | 5.2 (3.8–6.7) | |
| Sex | | | <0.001 | | <0.001 |
| Female | 11,032 (72.0%) | 11.5 (10.6–12.5) | | 7.1 (5.2–9.2) | |
| Male | 4239 (27.7%) | 11.6 (10.6–12.5) | | 8.3 (6.3–10.6) | |
| Unknown | 56 (0.4%) | 11.8 (10.7–12.5) | | 7.2 (4.8–9.4) | |
| Race | | | <0.001 | | <0.001 |
| Black | 803 (5.2%) | 12.2 (11.3–13.1) | | 6.9 (5.0–9.2) | |
| White | 12,212 (79.7%) | 11.8 (10.9–12.6) | | 7.4 (5.5–9.7) | |
| Other | 1555 (10.1%) | 11.5 (10.6–12.5) | | 7.3 (5.5–9.5) | |
| Unknown | 757 (4.9%) | 11.6 (10.6–12.4) | | 7.6 (5.8–9.6) | |
| Education | | | <0.001 | | <0.001 |
| College | 5572 (36.4%) | 11.6 (10.7–12.5) | | 7.9 (6.0–10.1) | |
| Some college | 3659 (23.9%) | 11.6 (10.3–12.4) | | 6.7 (4.8–8.9) | |
| No college | 10,001 (6.5%) | 11.4 (10.3–12.5) | | 6.8 (4.7–9.1) | |
| Unknown | 5095 (33.2%) | 11.5 (10.6–12.5) | | 7.5 (5.7–9.7) | |
| Alcohol (standard drinks) | | | <0.001 | | <0.001 |
| 4 or more per week | 2160 (14.1%) | 11.2 (10.3–12.2) | | 8.3 (6.4–10.6) | |
| 2 to 3 per week | 2487 (16.2%) | 11.3 (10.5–12.2) | | 8.1 (6.2–10.3) | |
| 2 to 4 per month | 3628 (23.7%) | 11.5 (10.6–12.4) | | 7.7 (5.9–9.8) | |
| Monthly or less | 4576 (29.9%) | 11.8 (10.8–12.8) | | 6.8 (4.9–8.9) | |
| Never | 1912 (12.5%) | 11.9 (10.7–12.8) | | 6.6 (4.5–9.2) | |
| Missing | 564 (3.7%) | 11.8 (10.8–12.7) | | 6.8 (4.9–9.3) | |
| Smoking | | | 0.863 | | <0.001 |
| Never | 9898 (64.6%) | 11.6 (10.7–12.5) | | 7.6 (5.7–9.8) | |
| Ever (≥100 cigarettes) | 5131 (33.5%) | 11.6 (10.6–12.6) | | 7.0 (5.1–9.4) | |
| Unknown | 298 (1.9%) | 11.5 (10.6–12.4) | | 7.4 (5.4–9.5) | |

[a]Values are reported as N (%). Percentages may not add up to 100% since patients can decline to answer survey questions.
[b]Values are reported as median (interquartile range).
[c]P-value in median sedentary time and daily steps across categories were derived from two-sided Kruskal–Wallis test or Wilcoxon test.

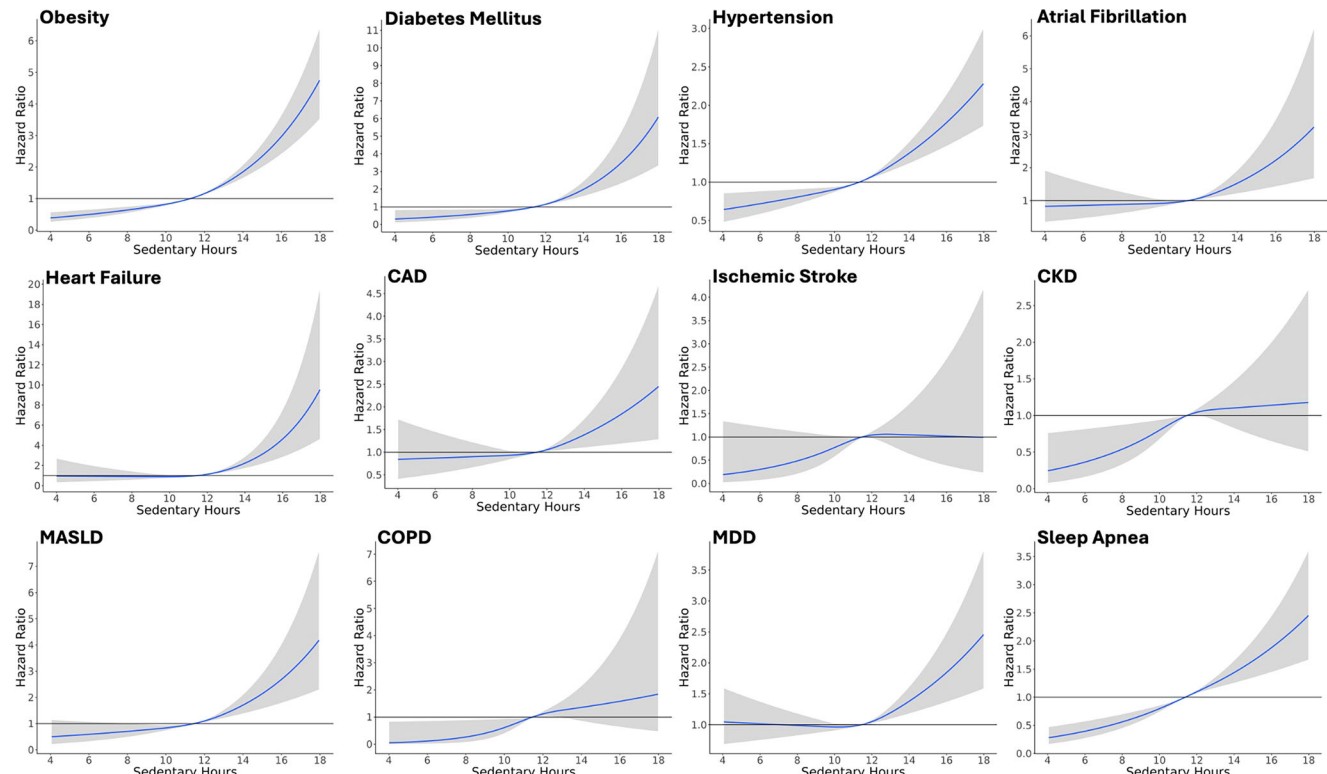

**Fig. 1 | Relationship between sedentary time and incident chronic disease.** Cox proportional hazard models were used to compute hazard ratios as a function of average daily sedentary time with the median sedentary time (11.7 h) used as reference. For each sedentary analysis, the blue line represents the hazard ratio, the gray area represents the 95th confidence interval, and the horizontal gray line indicates the hazard ratio of 1.00. All Cox proportional hazard models were adjusted for age, sex, smoking status, alcohol drinking status, and education status.

**Table 2 | Hazard ratios and 95% confidence intervals comparing 75th percentile to 25th percentile derived from time-varying Cox proportional hazard models including sedentary time only vs. sedentary time and steps**

| Chronic disease phenotype | Case/Control | Model with sedentary time | Model with sedentary time and steps | |
|---|---|---|---|---|
| | | Sedentary time HR (95% CI) | Sedentary time HR (95% CI) | Steps HR (95% CI) |
| Obesity | 1641/7608 | 1.45 (1.37–1.53) | 1.10 (1.04–1.17) | 0.46 (0.42–0.50) |
| Diabetes mellitus | 220/12,295 | 1.55 (1.36–1.78) | 1.28 (1.11–1.48) | 0.56 (0.45–0.70) |
| Hypertension | 2215/5771 | 1.21 (1.16–1.27) | 1.06 (1.01–1.12) | 0.70 (0.66–0.75) * |
| Atrial fibrillation | 232/12,758 | 1.25 (1.10–1.43) | 1.00 (0.87–1.16) | 0.52 (0.42–0.65) |
| Heart failure | 133/13,018 | 1.46 (1.26–1.70) * | 1.13 (0.95–1.34) | 0.46 (0.36–0.59) * |
| CAD | 442/12,416 | 1.18 (1.05–1.32) | 1.08 (0.94–1.24) | 0.75 (0.63–0.90) * |
| Ischemic Stroke | 99/13,063 | 1.22 (0.93–1.62) | 0.95 (0.70–1.30) | 0.53 (0.35–0.80) |
| CKD | 283/12,829 | 1.23 (1.04–1.44) | 1.07 (0.89–1.30) | 0.70 (0.56–0.87) * |
| MASLD | 553/12,957 | 1.40 (1.24–1.58) | 1.10 (0.96–1.26) | 0.51 (0.42–0.61) * |
| COPD | 50/12,748 | 1.66 (1.16–2.38) | 1.36 (0.93–1.98) | 0.60 (0.38–0.93) |
| MDD | 741/10,549 | 1.15 (1.07–1.23) * | 0.94 (0.86–1.02) | 0.55 (0.48–0.62) |
| Sleep Apnea | 849/11,236 | 1.41 (1.30–1.51) | 1.10 (1.01–1.20) | 0.53 (0.47–0.59) |

All Cox proportional hazards models were adjusted for age, sex, smoking status, alcohol drinking status, education status, and monthly averages of daily Fitbit wear time.

*HR* hazard ratio, *CI* confidence interval, *CAD* coronary artery disease, *CKD* chronic kidney disease, *MASLD* metabolic dysfunction-associated steatotic liver disease, *COPD* chronic obstructive pulmonary disease, *MDD* major depressive disorder.

*Significant nonlinear association between the variable and disease phenotype based on Wald $\chi^2$ tests.

significance (1.22; 0.93–1.62). When plotting hazard ratio as a function of sedentary time (Fig. 1), there was a consistent dose-response association between sedentary time and incidence for most of the chronic diseases. The exceptions were heart failure and MDD, where there was non-linearity in the association between sedentary time and incidence ($P$-value for non-linearity <0.05). For these two phenotypes, lower levels of sedentary time were not associated with reduced risk. In a sensitivity analysis using step cadence data to define sedentary

cadence time, we observed that higher sedentary cadence time was also associated with increased risk for 11 of the 12 chronic diseases, excluding ischemic stroke (Supplementary Table 1).

When both daily sedentary time and steps were included in the time-varying Cox proportional hazard models, sedentary time was significantly associated with increased risk of obesity (1.10; 1.04–1.17), diabetes mellitus (1.28; 1.11–1.48), hypertension (1.06; 1.01–1.12), and sleep apnea (1.10; 1.01–1.20) (Table 2). Higher daily steps was

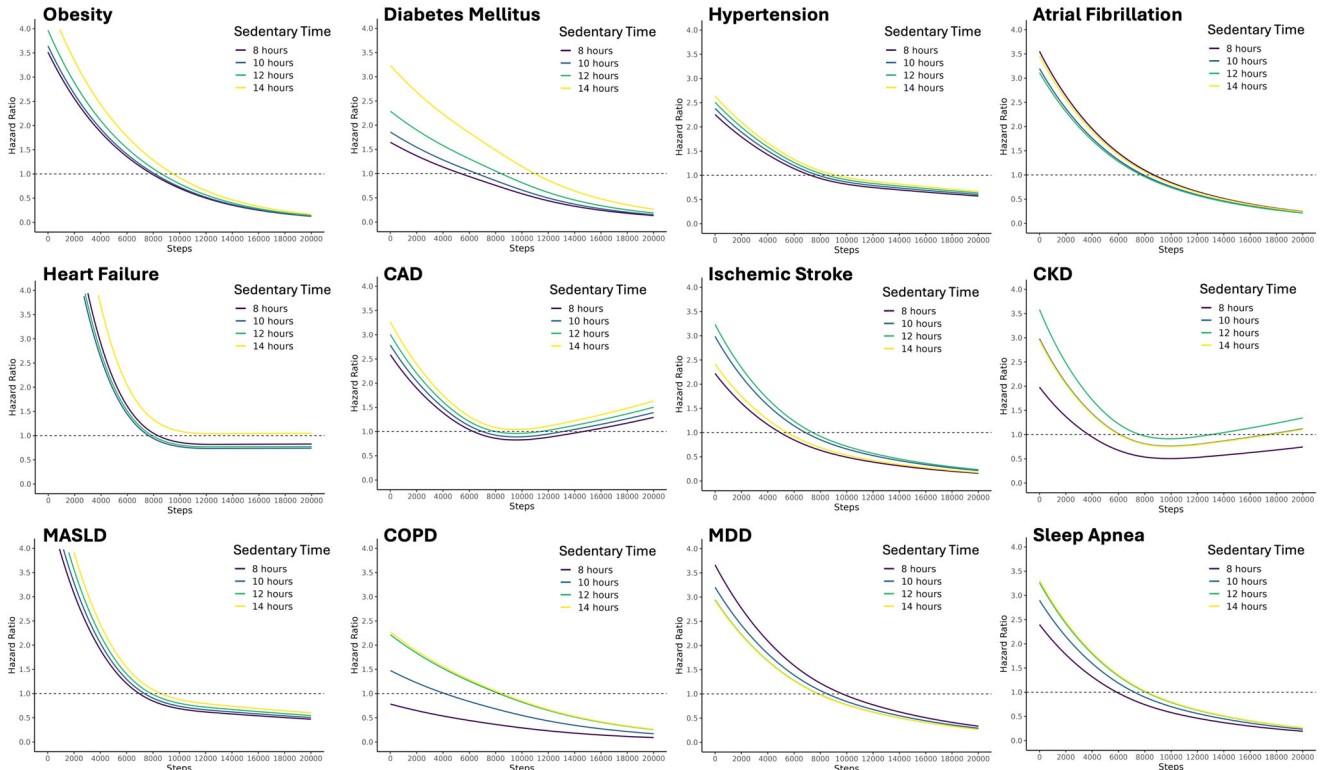

**Fig. 2 | Relationship between sedentary time, daily steps, and incidence of chronic disease.** Cox proportional hazard models were used to compute hazard ratios as a function of average daily steps and stratified by different levels of sedentary time. The cohort median sedentary time (11.7 h) and step count (7001) were used as the reference. The dotted horizontal line indicates a hazard ratio of 1.00. All Cox proportional hazard models were adjusted for age, sex, smoking status, alcohol drinking status, education status, and monthly averages of daily Fitbit wear time. CAD coronary artery disease, CKD chronic kidney disease, MASLD metabolic dysfunction-associated steatotic liver disease, COPD chronic obstructive pulmonary disease, MDD major depressive disorder.

associated with reduced risk for all 12 of the chronic disease, including obesity (0.46; 0.42–0.50), diabetes mellitus (0.56; 0.45–0.70), hypertension (0.70; 0.66–0.75), atrial fibrillation (0.52; 0.42–0.65), heart failure (0.46; 0.36–0.59), CAD (0.75; 0.63–0.90), ischemic stroke (0.53; 0.35–0.80), CKD (0.70; 0.56–0.87), MASLD (0.50; 0.42–0.61), COPD (0.60; 0.38–0.93), MDD (0.55; 0.48–0.62), and sleep apnea (0.53; 0.47–0.59). Non-linearity was observed in the association between daily steps and chronic disease incidence for hypertension, heart failure, CAD, and MASLD (Fig. 2). For hypertension, heart failure, and MASLD, the risk reduction plateaus after approximately 8000 daily steps. For CAD, there is a J-shaped association with risk increasing above 12,000 daily steps and exceeding hazard ratio of 1.00 at >16,000 steps. Notably, for both heart failure and CAD, the hazard ratio did not reach 1.00 or lower for patients with 14 h of daily sedentary time at any daily step count between 0 and 20,000 steps. We also estimated cumulative 3-year incidence for the 12 chronic diseases at different levels of daily sedentary time and step count (Supplementary Fig. 2), which showed trends similar to those from the hazard ratio plots.

Using a bootstrapping approach, we estimated the daily steps required to mitigate the risk of incident chronic disease associated with sedentary behavior (Fig. 3). Compared to the cohort median, individuals with greater sedentary time (14 h vs. 8 h) required more steps to offset risk for obesity (1700 daily steps), MASLD (1700), hypertension (2200), sleep apnea (2200), diabetes mellitus (5300), and COPD (5500). For obesity, the number of steps needed to offset risk of incident disease increased monotonically with baseline BMI when stratifying by baseline BMI. For CAD, no number of daily steps could offset the risk of 14 h of sedentary time, but there was an upward trend in number of daily steps needed to offset increasing sedentary time for those with fewer than 14 h of sedentary time. The number of steps required to offset risk of heart failure was similar between

different levels of sedentary time. Notably, there was an inverse trend for MDD, where individuals with 14 h of sedentary time required 1800 fewer steps than those with 8 h. For ischemic stroke and chronic kidney disease (CKD) individuals with 14 h of sedentary time required fewer steps than those with 12 h to mitigate risk. For atrial fibrillation, individuals at both extremes (8 or 14 h of sedentary time) required more steps to reduce risk than those with 10 or 12 h. Case counts were low at the extremes of sedentary behavior (8 or 14 h of sedentary time) for these phenotypes.

## Discussion

In this study, we leveraged longitudinal, continuous measurements of sedentary time and step count over many years from commercial wearable devices linked to EHRs to investigate the relationship between sedentary time, daily steps, and incident chronic disease. We demonstrate that higher sedentary time is consistently associated with increased risk of multiple chronic diseases, whereas higher daily steps appear to have a protective effect. We quantified the number of steps required to offset the risk associated with sedentary behavior, which varied depending on baseline sedentary time and the specific chronic disease. These findings suggest that certain chronic diseases may be more susceptible to the impact of sedentary time and daily steps compared to others.

Our study design and data sources differ from prior work in several important ways. To our knowledge, this is the first study to use years of sedentary time directly measured from commercial wearable devices, totaling over 13 million days of observation. Prior studies have relied on self-reported measures or short-term accelerometry, which cannot capture seasonal or individual variation in sedentary behavior and are prone to bias[4,9,22–24]. We account for longitudinal behavioral changes by including time-varying monthly estimates of sedentary

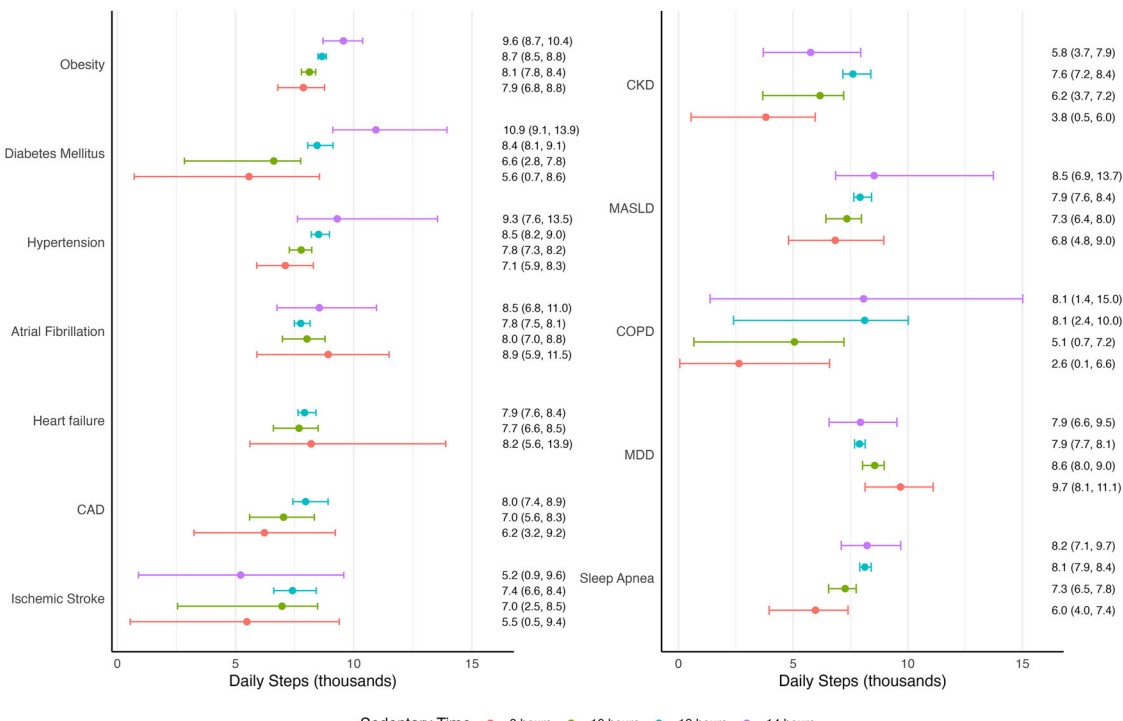

**Fig. 3 | Daily steps required to offset the risk of sedentary behavior on chronic disease.** Daily step estimates were derived from 1000 bootstrap iterations of resampling and refitting of the Cox proportional hazard models. For each analysis, the point represents the median daily steps, the error bars represent the 95th confidence interval, and the corresponding description is included to the right. The Cox proportional hazard models were adjusted for age, sex, smoking status, alcohol drinking status, education status, and monthly averages of daily Fitbit wear time. CAD coronary artery disease, CKD chronic kidney disease, MASLD metabolic dysfunction-associated steatotic liver disease, COPD chronic obstructive pulmonary disease, MDD major depressive disorder.

time, step count, and device wear time in our models. Unlike most prior studies that focused on activity minutes, we evaluated daily step count, which is a practical and accessible marker. Our findings are especially relevant for individuals monitoring their daily steps and sedentary behavior and enable healthcare providers to provide tailored recommendations. Additionally, most literature on physical activity has focused on one of cardiovascular disease, diabetes, cancer, or mortality. In this study, we assessed multiple disease phenotypes simultaneously by integrating wearable and EHR data.

The median sedentary time in our cohort was 11.6 h per day, which is greater than estimates from prior studies using survey or short-term accelerometry data (7.7–10.6 h)[2,13,25,26]. Our longer observation window provides a more reliable estimate of long-term behavior. Therefore, it is likely that U.S. adults are more sedentary than previously estimated, especially since self-reported measures or short-term accelerometry are susceptible to bias[9,22,23]. Higher sedentary time was observed among older adults, Black participants, and those without college degrees, which aligns with prior findings[27,28]. These populations with higher risk sedentary behavior may benefit from targeted recommendations, enhanced clinical monitoring, and further research aimed at improving health outcomes.

Our time-varying analyzes showed that increased sedentary time is consistently associated with greater risk of chronic diseases across cardiovascular, pulmonary, metabolic, liver, kidney, and psychiatric conditions. Reduction of sedentary time may be protective against incident chronic disease, such as obesity, diabetes mellitus, hypertension, CKD, MASLD, and sleep apnea. For most conditions, there appeared to be a dose-response association between daily sedentary time and risk of chronic disease. These findings add to the growing body of literature showing that sedentary behavior is a cross-system risk factor for chronic diseases[1–6,9,29–32]. Several proposed pathways could explain these associations, including cardiorespiratory fitness, energy metabolism, musculoskeletal mass, immune suppression, and cerebral blood flow[5,31].

Increased daily steps were consistently associated with lower risk for all 12 chronic diseases[15,18,33]. However, the steps required to offset sedentary behavior varied. For example, compared to 8 h of sedentary time, individuals with 14 h needed an additional 5300 steps to offset the risk for diabetes mellitus, but only 1700 steps for obesity. This variation suggests that daily steps and sedentary behavior have different impacts across diseases. We observed non-linear associations between step count and several conditions. For hypertension, heart failure, and MASLD, risk reduction plateaued around 8000 steps. These findings support 8000 steps as a practical target for health benefits, which is supported by recent studies that found between 7000 and 9000 daily steps is sufficient for most adults[13–16,34,35].

We observed a J-shaped association between daily steps and CAD, with risk increasing above 12,000 daily steps and exceeding baseline risk at >16,000 steps. This mirrors findings from the UK Biobank, which showed a similar trend of increased cardiovascular disease incidence at higher levels of daily steps[13]. One hypothesis is that long-term excessive physical activity, such as endurance exercise, may induce adverse cardiovascular remodeling[36]. These results suggest an upper bound to the cardiovascular benefits of physical activity.

Importantly, for heart failure and CAD, the hazard ratio for patients with 14 h of sedentary time never reached 1.00 at any step count, suggesting that extreme sedentary time confers risk that may not be fully reversible by daily steps. Similarly, a recent UK Biobank found that individuals in the highest quartile of sedentary time who met MVPA guidelines still had a 33% higher risk of cardiovascular mortality[2]. These findings support the growing literature that suggests sedentary behavior may be, at least in part, independent of physical activity with regards to heart failure and CAD risk.

Some of our findings were unexpected or counterintuitive but may have rational explanations. For heart failure and MDD, lower sedentary time was not associated with reduced risk. In these instances, increased sedentary time may be an early behavioral indicator of disease (e.g., functional limitations in heart failure or psychomotor slowing in MDD) rather than a causal factor[37]. We also noted an inverse relationship between daily steps and sedentary time for MDD. Higher sedentary levels in patients with MDD may reflect psychomotor slowing[38]. Patients with psychomotor slowing may show greater responsiveness to increased daily steps[39]. Other counterintuitive patterns (e.g., fewer steps needed to offset risk at 14 vs 12 sedentary hours for CKD and stroke) occurred in subgroups with small sample sizes ($N < 50$) and require further validation.

There are several limitations to this study. First, sedentary time was derived from Fitbit devices and is based on a proprietary algorithm. While Fitbit-derived sedentary time has been validated in both younger and older adults, potential for systematic misestimation remains[40–42]. Additionally, participants were not restricted to a single model of Fitbit devices, but the Fitbit algorithm has not been changed since launching in 2017[19]. Notably, our sensitivity analysis using sedentary cadence time derived directly from step cadence data showed similar results to the primary analysis with Fitbit-derived sedentary time, reinforcing the validity of Fitbit-derived sedentary time. Second, the study cohort is relatively young, majority female, White, and college-educated, which may limit the generalizability of the findings to underrepresented populations. There is also a lower prevalence of some common chronic diseases, such as diabetes, compared to the general population. Therefore, point estimates may be unstable due to low case count, especially at the extremes of sedentary behavior (8 or 14 h of daily sedentary time). Nonetheless, our findings are consistent with those from studies using survey or accelerometry data, highlighting that Fitbit devices linked to EHRs capture fundamental relationships between sedentary behaviors and chronic disease, even if absolute estimates differ. Third, our sedentary time estimates lack behavioral context and cannot distinguish between patterns such as intermittent breaks versus prolonged sedentary bouts, which may carry different risks[4]. Likewise, step count does not capture the intensity or quality of physical activity. However, prior studies have shown that step count has more effect on mortality than step intensity[34]. Fourth, this was an EHR-registry-based study, and data were not available for family history, dietary habits, and medication history, which may have introduced confounding in the associations between sedentary behavior and chronic disease risk. Future prospective cohort studies may help elucidate these nuanced relationships. Fifth, the possibility of reverse causation cannot be fully removed from retrospective, observational data. We attempted to mitigate this risk by focusing on incident diagnoses occurring at least 6-months after Fitbit monitoring began. Finally, we acknowledge the potential for misclassification of outcomes when using EHR data.

In summary, sedentary behavior is associated with a higher incidence of numerous chronic diseases, and increased daily steps can offset this risk. The risk reduction associated with daily steps varied based on an individual's baseline sedentary behavior and the specific chronic disease. These findings add to a growing evidence base that supports incorporating quantitative daily activity recommendations for both sedentary time and daily steps. Furthermore, our findings, together with future research in underrepresented populations and on unmeasured factors such as diet, could help healthcare providers deliver personalized, evidence-based recommendations tailored to individual activity patterns and chronic disease risk.

## Method

### Study participants
All study participants consented and enrolled in the All of Us research program from May 2017 to October 2023, as described previously[17,43].

We used data from the All of Us Controlled Tier Dataset v8 (C2024Q3R4), which is a retrospective electronic health record-based database available on the All of Us Researcher Workbench. From over 633,000 All of Us participants enrolled at the time of analysis, 58,527 adult (≥18 years) participants voluntarily shared Fitbit device data through the Bring Your Own Device program or the Wearables Enhancing All of Us Research (WEAR) Study[17,43–45]. We excluded patients without Fitbit-derived sleep data to distinguish between sedentary time and sleep time.

### Fitbit activity data
We used consumer-facing Fitbit activity metrics, including daily sedentary time and daily step count. Fitbit estimates sedentary time using a proprietary algorithm based on heart rate and movement. Baseline daily sedentary time and step count were calculated as the average over the initial 6-months of monitoring. Monthly averages of daily sedentary time and step count after this period were calculated as time-varying variables. We excluded months with ≤15 valid days (valid days: ≥10 h of wear time, 100 to 45,000 steps, and 0 to 1440 min of sedentary time)[15,19].

### Study outcomes and covariates
Primary outcomes for our analyzes were incident diagnoses of 12 common, chronic conditions previously linked to physical activity behavior[31]: obesity, diabetes mellitus, hypertension, atrial fibrillation, heart failure, coronary artery disease (CAD), ischemic stroke, chronic kidney disease (CKD), metabolic dysfunction-associated steatotic liver disease (MASLD), chronic obstructive pulmonary disease (COPD), major depressive disorder (MDD), and sleep apnea. The selected chronic diseases were identified using a combination of International Classification of Diseases (ICD) billing codes, procedure codes, medications, and vitals measurements (Supplementary Table 2). We used algorithms for validated EHR phenotypes when available[46], and direct vitals measurements for obesity and hypertension. Covariates derived from enrollment surveys included age, sex, self-reported race, education, smoking status, and alcohol intake.

### Statistical analysis
We calculated median daily sedentary time and step count for the patient cohort stratified by demographic and lifestyle characteristics. Differences across categories were tested using the Kruskall–Wallis test or Wilcoxon test.

We applied Cox proportional hazards models to analyze the association between time-varying monthly estimates of sedentary time and the 12 selected chronic diseases. The entry date was the participant's first date of Fitbit monitoring. Participants were censored at their last medical encounter, which can extend beyond the last date of Fitbit monitoring. For each chronic disease of interest, we excluded patients with prevalent diagnoses or incident diagnoses within the first 6-months of Fitbit monitoring to mitigate the risk of reverse causation. Hazard ratios and 95% confidence intervals were calculated comparing the 75th and 25th percentile of sedentary time. We estimated hazard ratios as a function of sedentary time relative to the cohort median. All models were adjusted for age, sex, smoking status, alcohol drinking status, education status, and monthly averages of daily Fitbit wear time. Continuous variables such as age and sedentary time were modeled as restricted cubic splines with three knots. Wald $\chi^2$ tests were used to assess for nonlinearity.

We then modeled monthly estimates of sedentary time and step count jointly using the same Cox proportional hazards approach. We estimated the number of steps needed to offset sedentary risk (i.e., achieve a hazard ratio = 1.00) for individuals averaging 8, 10, 12, or 14 h of sedentary time per day compared to the cohort median. Step estimates and corresponding 95% confidence intervals were derived from

1000 bootstrap iterations, refitting the Cox model on each resampled dataset.

All statistical tests were based on two-tailed probability. Statistical analyzes were performed in R (version 4.5.0, R Project https://www.r-project.org).

## Sensitivity analysis with step cadence

We calculated the number of minutes per day that each participant spends in a sedentary cadence. Sedentary cadence was defined as non-movement (0 step/min with heart rate data for that minute) or incidental movement (1–19 steps/min). We then applied Cox proportional hazards models to analyze the association between time-varying monthly estimates of sedentary cadence time and step count with the 12 selected chronic diseases with the same methodology as the primary analysis.

## Reporting summary

Further information on research design is available in the Nature Portfolio Reporting Summary linked to this article.

## Data availability

To ensure privacy of participants, data used for this study are available to approved researchers following registration, completion of ethics training and attestation of a data use agreement through the All of Us Research Workbench platform, which can be accessed via https://workbench.researchallofus.org/. Source data for the hazard ratio curves and 95th confidence intervals in Figs. 1 and 2 are provided with this paper. Source data are provided with this paper.

## Code availability

Code used for this study can be found at https://github.com/nszheng/aou_fitbit_sedentary. Evaluating the code requires access to the All of Us Research Workbench platform, which is available to approved researchers following registration, completion of ethics training and attestation of a data use agreement. Code can also be made available to any approved researchers on the All of Us Research Workbench platform by contacting our study team.

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

## Acknowledgements

The study was supported by National Institute of Health grants R21 HL172038 (E.L.B.), R61/R33 HL158941 (E.L.B.), and R01 FD007627 (E.L.B.). We gratefully acknowledge All of Us participants for their contributions, without. whom this research would not have been possible. We also thank the National. Institutes of Health's All of Us Research Program for making available the cohort. examined in this study. The All of Us Research Program is supported by the National Institutes of Health, Office of the Director: Regional Medical Centers (1 OT2 OD026549; 1 OT2 OD026554; 1 OT2 OD026557; 1 OT2 OD026556; 1 OT2 OD026550; 1 OT2 OD 026552; 1 OT2 OD026553; 1 OT2 OD026548; 1 OT2 OD026551; 1 OT2 OD026555; IAA: AOD21037, AOD22003, AOD16037, AOD21041), Federally Qualified Health Centers (HHSN 263201600085U), Data and Research Center (5 U2C OD023196), Biobank (1 U24 OD023121), The Participant Center (U24 OD023176), Participant Technology Systems Center (1 U24 OD023163), Communications and Engagement (3 OT2 OD023205; 3 OT2 OD023206) and Community Partners (1 OT2 OD025277; 3 OT2 OD025315; 1 OT2 OD025337; 1 OT2 OD025276).

## Author contributions

N.S.Z. and E.L.B. conceived of and designed the study. N.S.Z. and J.A. performed with data extraction and cleaning. N.S.Z. performed statistical analyses with assistance from S.H. H.M. and K.M.F. assisted with data interpretation. N.S.Z. and E.L.B. drafted the manuscript with participation from all authors. All authors reviewed and approved the final manuscript.

## Competing interests

E.L.B has received research funds unrelated to this work from United Therapeutics and Anumana and unrestricted funds for research from Google. The All of Us Research Program was not involved in the design and conduct of the study; collection, management, and analysis of the data; preparation, review, or approval of the manuscript; and decision to submit the manuscript for publication. To ensure privacy of participants, data used for this study was accessed and available to approved researchers only following registration, completion of ethics training, and attestation of a data use agreement through the All of Us Research Workbench platform, which can be accessed via https://workbench.researchallofus.org/login. The remaining authors declare no competing interests.
