## [Transparent Peer Review file · Nature Communications]

Daily Steps Offset Risks of Sedentary Behavior in the All of Us Research Program

Corresponding Author: Dr Evan Brittain

Version 0:

Reviewer comments:

Reviewer #1

(Remarks to the Author)

Thank you for the opportunity to review the present manuscript that investigates whether increasing daily step count can offset the risks of 12 chronic conditions associated with sedentary behavior. The study is based on retrospective case-control studies utilizing continuous longitudinal data on sedentary time and step count from Fitbit devices linked to electronic health records in a large study sample of adults. The manuscript is well-written overall and addresses an interesting topic with potential clinical and public health implications. However, there are several issues that need attention according to the following:

- The introduction should acknowledge previous research investigating potential interaction effects between sedentary behaviors and step counts on disease risk. For example, the authors could highlight the work by Ahmadi et al. (reference 12) who reported data about this and further explain how the current manuscript would advance knowledge about inter-relationships between sedentary behaviors and step counts for disease risk.
- The introduction currently lacks a clear rationale behind investigating sedentary time and step counts specifically, thus without acknowledging information about physical activity intensity or types of activities. The authors claim that this approach may support more intuitive and accessible physical activity recommendations. However, it remains unclear what research gap that the current manuscript is addressing.
- The description of the study design is unclear. It is stated in the abstract that the study is based on case-control studies, with no further description in the method section of the manuscript. Please add more information about the study design, including study duration.
- A clear operational definition of sedentary behavior is currently lacking in the manuscript. For example, does time spent sedentary imply that no steps are recorded or can you accumulate steps and still be classified as being sedentary? If you can accumulate steps while being classified as sedentary, how many steps can be accumulated during a given time window? This is important to clarify in order to understand the relationship between sedentary time and step counts and thereby the interpretation of the results.
- The analyses lack important covariates for disease risk, including parental disease history, medication use and information about dietary habits. The latter lifestyle behavior also involves snacking behaviors, which typically occurs during sedentary behaviors. For example, excess of adiposity and related morbidities may be due to sedentary behaviors per se or the intake of energy-dense foods while being sedentary (or a joint combination). Either way, dietary habits need to be considered in order to determine the true impact of sedentary behaviors on disease outcomes.
- In lines 297 – 298, the authors states that their results “.....suggests that physical activity and sedentary behavior have different impacts across diseases”. Although step counts are recorded as a result of physical activity, many exercise activities are being excluded when defining physical activity in terms of step counts. For example, most resistance-type exercises do not add much to the total accumulated number of steps, nor do bicycling, swimming or yoga. Given that a large set of exercises cannot be properly assessed with the Fitbit monitor, it may be more appropriate to replace the term physical activity with step counts throughout the manuscript to clarify what was actually assessed in this study.

- The authors highlight the results that individuals with greater sedentary time (14 hours vs. 8 hours) required more steps to offset risk for several disease states (also highlighted in the abstract). However, the 95% confidence intervals for these point estimates are generally very wide. For example, the point estimate for the 8-hour category regarding COPD is reported to be somewhere between 100 and 6600 steps, with the corresponding interval for the 14-hour category to be between 1400 and 15000 steps. Although the authors do acknowledge that point estimates at the extremes of sedentary behaviors may be unstable due to low case count, 95% confidence intervals remain wide also for less extreme sedentary behaviors. For example, the point estimate for the 10-hour category regarding diabetes mellitus risk was reported to be found somewhere between 2800 and 7800 steps. Given these wide 95% confidence intervals, revealing a low precision of reported point estimates, it may be questioned to what extent these results may help healthcare providers in delivering more personalized, evidence-based recommendations, as the authors claim in their conclusions.

(Remarks on code availability)

Reviewer #2

(Remarks to the Author)

Thank you for the opportunity to review this study . The manuscript is well written and analyses a large data set of wearable physical activity (step count) and sedentary behaviour data. A strength is the inclusion of a range of chronic disease health outcomes and the use of longitudinal behavioural data.

Please see below for suggestions to further improve the manuscript:

Abstract

1. Line 46: It would be beneficial to name Fitbit as a wearable activity tracker/device (or something similar) for clarity as some readers may not be aware of specific brands

Introduction

2. General: It would be beneficial if the importance of steps/daily step count to health is briefly introduced. For example, evidence from recent research such as 'Daily steps and health outcomes in adults: a systematic review and dose-response meta-analysis, DOI:10.1016/S2468-2667(25)00164-1'
3. Line 78: It may be beneficial to briefly define SB and PA to ensure all readers are familiar with key behaviours explored in the study
4. Line 88: For improved clarity it would be worthwhile to highlight that whilst specific quantitative guidance for SB is not included, the message is to reduce time spent in this behaviour and replace with PA.
5. Lines 90-9: Can the authors provide any data to support the 'widespread use of commercial devices and smartphones'? For example, are there any data that can be included reported the number of devices used/bought/worn etc.

Methods

6. Fitbit data: Were only certain models/versions of Fitbit included in the data set or were different models permitted? If different models/versions are included, is there the possibility that algorithms for calculating activity metrics will have evolved over time? If this, this would need to be noted as a limitation
7. Line 127-128: What was the justification for these criteria for a valid day? Can a reference/rationale be provided in the manuscript to support this?

Table 1

8. For the age categories, the cross over between categories makes it unclear how participants were grouped. For example, if someone was aged 40 years old, were they in the 18-40 or 40-60 category?
9. For clarity, for alcohol classifications it would be beneficial to specify if the numerical values are referring to units or number of drinks.

Discussion

10. General: It is concluded that 'results may help healthcare providers deliver more personalized, evidence-based recommendations tailored to individual behavior patterns and chronic diseases'. Given the variation in step count totals that were required to offset SB for different diseases, have the authors considered how practical this could be? Research has suggested that healthcare providers often do not know the global PA guidelines (i.e., 150mins per week), therefore, how feasible would it be for them to be recommending bespoke step counts for different SB profiles and health conditions? It would be beneficial to consider this further in the manuscript
11. Lines 252: Please amend typo 'that that'
12. Line 301: It would be beneficial to discuss this finding in relation to recent work suggesting 7,000 steps is sufficient for health outcomes (Daily steps and health outcomes in adults: a systematic review and dose-response meta-analysis, DOI:10.1016/S2468-2667(25)00164-1).

(Remarks on code availability)

Version 1:

Reviewer comments:

Reviewer #1

(Remarks to the Author)

I congratulate the authors for successfully improving the quality of the manuscript. Importantly, the authors have added a paragraph acknowledging a limited generalizability of the findings to underrepresented populations (the study cohort being relatively young, majority female, White, and college-educated), together with unstable point estimates of disease risk reductions due to low case count, and that results may be confounded by important unmeasured covariates such as dietary habits. Given these limitations, to what extent the results "may help healthcare providers to deliver more personalized, evidence-based recommendations tailored to individual behavior patterns and chronic disease risk profiles" may be questioned. I recommend the authors to consider a revision of the conclusion of the manuscript highlighting the need for further investigations in order to develop personalized evidence-based physical activity recommendations.

(Remarks on code availability)

Reviewer #2

(Remarks to the Author)

Thank you to the authors for amending their manuscript in response to reviewer feedback. All points raised have been satisfactorily addressed.

(Remarks on code availability)

Not applicable

In cases where reviewers are anonymous, credit should be given to 'Anonymous Referee' and the source. The images or other third party material in this Peer Review File are included in the article's Creative Commons license, unless indicated otherwise in a credit line to the material. If material is not included in the article's Creative Commons

February 3, 2026
Nature Communications

Department of Medicine
Division of Cardiovascular Medicine

RE: Daily Steps Offset Risks of Sedentary Behavior for Chronic Diseases: Insights from the All of Us Research Program

We thank the reviewers for taking their time to review our work and for the opportunity to revise the manuscript. We have revised the manuscript to address their comments and to follow the *Nature Communications* publication style. Our modifications are listed below and are highlighted in yellow in the revised manuscript.

Comments from the Reviewer #1:

Thank you for the opportunity to review the present manuscript that investigates whether increasing daily step count can offset the risks of 12 chronic conditions associated with sedentary behavior. The study is based on retrospective case-control studies utilizing continuous longitudinal data on sedentary time and step count from Fitbit devices linked to electronic health records in a large study sample of adults. The manuscript is well-written overall and addresses an interesting topic with potential clinical and public health implications. However, there are several issues that need attention according to the following:

- 1) The introduction should acknowledge previous research investigating potential interaction effects between sedentary behaviors and step counts on disease risk. For example, the authors could highlight the work by Ahmadi et al. (reference 12) who reported data about this and further explain how the current manuscript would advance knowledge about inter-relationships between sedentary behaviors and step counts for disease risk.**

Thank you for the suggestion. We have highlighted the work by Ahmadi et al. and added additional text explain current limitations in our knowledge on the relationship between sedentary behavior and step count. Specifically, prior studies, including Ahmadi et. al, have relied on short duration actigraphy which cannot capture natural, longitudinal behavior. Furthermore, prior studies have focused on primary cardiovascular disease and mortality, which excludes other important chronic diseases such as obesity, diabetes, and hypertension.

Corresponding text has been added to the Introduction on page 3:

“Recently, Ahmadi et. al reported that roughly 9,000 to 10,500 steps per day was associated with the lowest risk of incident cardiovascular disease or mortality independent of sedentary behavior using actigraphy data collected over two weeks in the UK Biobank.¹³ However, shorter duration actigraphy-based measures of daily count and sedentary behavior cannot capture longitudinal, natural behavior due to seasonal or individual variations as well as potential observer bias. Additionally, prior studies have focused on mortality and cardiovascular disease and there have been limited reports on the joint effect of daily steps and sedentary behavior on other common chronic diseases, such as obesity, type 2 diabetes mellitus, and hypertension.¹³”

- 2) The introduction currently lacks a clear rationale behind investigating sedentary time and step counts specifically, thus without acknowledging information about physical activity intensity or types of activities. The authors claim that this approach may support more intuitive and accessible physical activity recommendations. However, it remains unclear what research gap that the current manuscript is addressing.**

We appreciate the opportunity to clarify our study rationale. We believe that step count is a more accessible and easier to understand measure of physical activity compared to moderate-vigorous activity, especially now that more people own commercial wearable devices and smartphones that can report step count. However, there are currently no guidelines on healthy levels of daily steps and sedentary behavior. This study provides further evidence towards more personalized step count recommendations that may differ by sedentary behavior.

We've added corresponding text to the Introduction on page 3:

“U.S. guidelines for physical activity recommend MVPA (≥ 150 minutes per week) but offer no quantitative guidance on sedentary time or daily steps.¹⁰ The increasingly widespread use of commercial wearable devices and smartphones that can track step count allows patients to monitor step count as a more accessible measure of physical activity.¹¹ Several studies have suggested that daily steps and MVPA are highly correlated and may have similar associations with mortality and cardiovascular disease.¹²⁻¹⁵ These studies have typically recommended between 7,000 to 9,000 daily steps for the general population to reduce risk of chronic disease and mortality.¹²⁻¹⁵ However, it remains unclear whether individuals with greater sedentary behavior need additional steps to offset sedentary-associated risks for chronic diseases.”

- 3) The description of the study design is unclear. It is stated in the abstract that the study is based on case-control studies, with no further description in the method section of the manuscript. Please add more information about the study design, including study duration.**

Thank you for the opportunity to clarify the study design. This is a retrospective cohort study, and it is not a case-control study. This has been removed from the abstract. All study participants were enrolled between May 2018 to October 2023.

Corresponding text has been added to the Methods on page 18:

“All study participants consented and enrolled in the *All of Us* research program from May 2018 to October 2023, as described previously.^{16,42} We used data from the *All of Us* Controlled Tier Dataset v8 (C2024Q3R4), which is a retrospective electronic health record-based database available on the *All of Us* Researcher Workbench.”

- 4) A clear operational definition of sedentary behavior is currently lacking in the manuscript. For example, does time spent sedentary imply that no steps are recorded or can you accumulate steps and still be classified as being sedentary? If you can accumulate steps while being classified as sedentary, how many steps can be accumulated during a given time window? This is important to clarify in**

order to understand the relationship between sedentary time and step counts and thereby the interpretation of the results.

Thank you for this important point. Fitbit-derived sedentary time is a proprietary algorithm, and we do not have access to the underlying algorithm. However, Fitbit-derived sedentary time has been validated in prior studies against accelerometry. We have also added a sensitivity analysis that calculates sedentary cadence time directly from minute-to-minute step cadence data. The sensitivity analysis showed that results were consistent between Fitbit-derived sedentary time and our own calculated sedentary cadence time.

Corresponding text has been included in the Results on page 4 along with a new supplementary table (eTable 1):

“In a sensitivity analysis using step cadence data to define sedentary cadence time, we observed that higher sedentary cadence time was also associated with increased risk for 11 of the 12 chronic diseases, excluding ischemic stroke (eTable 1).”

We have added additional text to the limitations in the Discussion on page 7:

“First, sedentary time was derived from Fitbit devices and is based on a proprietary algorithm. While Fitbit-derived sedentary time has been validated in both younger and older adults, potential for systematic misestimation remains. Nonetheless, our sensitivity analysis using sedentary cadence time derived directly from step cadence data showed similar results to the primary analysis with Fitbit-derived sedentary time, reinforcing the validity of Fitbit-derived sedentary time.”

- 5) The analyses lack important covariates for disease risk, including parental disease history, medication use and information about dietary habits. The latter lifestyle behavior also involves snacking behaviors, which typically occurs during sedentary behaviors. For example, excess of adiposity and related morbidities may be due to sedentary behaviors per se or the intake of energy-dense foods while being sedentary (or a joint combination). Either way, dietary habits need to be considered in order to determine the true impact of sedentary behaviors on disease outcomes.**

We agree that covariates such as parental disease history, medication use, and dietary habits would be helpful with regards to fully understanding the impact of sedentary behavior on chronic disease outcomes. Unfortunately, this was an EHR-based study and not a prospective cohort study, so we do not have these covariates available for analysis.

We have added additional text in the Discussion to acknowledge these limitations on page 8:

“Fourth, this was an EHR-registry based study and data was not available for family history, dietary habits, and medication history, which may have introduced confounding in the associations between sedentary behavior and chronic disease risk. Future prospective cohort studies may help elucidate these nuanced relationships.”

- 6) In lines 297 – 298, the authors states that their results “.....suggests that physical activity and sedentary behavior have different impacts across diseases”. Although step counts are recorded as a result of physical activity, many exercise**

activities are being excluded when defining physical activity in terms of step counts. For example, most resistance-type exercises do not add much to the total accumulated number of steps, nor do bicycling, swimming or yoga. Given that a large set of exercises cannot be properly assessed with the Fitbit monitor, it may be more appropriate to replace the term physical activity with step counts throughout the manuscript to clarify what was actually assessed in this study.

Thank you for this suggestion. We have replaced the phrase physical activity with daily steps where appropriate throughout the manuscript.

- 7) **The authors highlight the results that individuals with greater sedentary time (14 hours vs. 8 hours) required more steps to offset risk for several disease states (also highlighted in the abstract). However, the 95% confidence intervals for these point estimates are generally very wide. For example, the point estimate for the 8-hour category regarding COPD is reported to be somewhere between 100 and 6600 steps, with the corresponding interval for the 14-hour category to be between 1400 and 15000 steps. Although the authors do acknowledge that point estimates at the extremes of sedentary behaviors may be unstable due to low case count, 95% confidence intervals remain wide also for less extreme sedentary behaviors. For example, the point estimate for the 10-hour category regarding diabetes mellitus risk was reported to be found somewhere between 2800 and 7800 steps. Given these wide 95% confidence intervals, revealing a low precision of reported point estimates, it may be questioned to what extent these results may help healthcare providers in delivering more personalized, evidence-based recommendations, as the authors claim in their conclusions.**

We agree with the reviewer that the large confidence intervals for several of the chronic disease phenotypes are difficult to interpret, especially for those with lower case count. The Fitbit participants are overall younger and healthier than the general population. Therefore, there are relatively lower case counts of diabetes in our study which contributes to the larger confidence intervals.

We have added additional text to the limitations in the Discussion on page 8: "Additionally, the study cohort is relatively young, majority female, White, and college-educated, which may limit generalizability of the findings to underrepresented populations. There is also lower prevalence of some common chronic diseases, such as diabetes, compared to the general population. Therefore, point estimates may be unstable due to low case count, especially at the extremes of sedentary behavior (8 or 14 hours of daily sedentary time)."

Comments from the Reviewer #2:

Thank you for the opportunity to review this study. The manuscript is well written and analyses a large data set of wearable physical activity (step count) and sedentary behavior data. A strength is the inclusion of a range of chronic disease health outcomes and the use of longitudinal behavioral data. Please see below for suggestions to further improve the manuscript:

- 1) Line 46: It would be beneficial to name Fitbit as a wearable activity tracker/device (or something similar) for clarity as some readers may not be aware of specific brands**

Thank you for the suggestion. We have edited the Introduction to add a description of Fitbit devices on page 3:

“Participants share multiple longitudinal sources of health-related information, including electronic health records (EHRs), physical measures, and data from Fitbit devices, which are commercial wearable activity trackers that provide objective longitudinal measurements of sedentary time and daily steps from participants.”

- 2) General: It would be beneficial if the important of steps/daily step count to health is briefly introduced. For example, evidence from recent research such as ‘Daily steps and health outcomes in adults: a systematic review and dose-response meta-analysis, DOI:10.1016/S2468-2667(25)00164-1’**

Thank you for the suggestion. We have added the suggested citation and the additional text to the Introduction regarding the importance of daily steps on page 3:

“Several studies have suggested that daily steps and MVPA are highly correlated and may have similar associations with mortality and cardiovascular disease.¹²⁻¹⁶ These studies have typically recommended between 7,000 to 9,000 daily steps for the general population to reduce risk of chronic disease and mortality.^{2,12-15} “

- 3) Line 78: It may be beneficial to briefly define SB and PA to ensure all readers are familiar with key behaviors explored in the study**

We have added definition for sedentary behavior in the Introduction on page 3:

“Sedentary behavior is defined as any waking behavior with low energy expenditure (≤ 1.5 metabolic equivalents), such as sitting or reclining.^{4”}

- 4) Line 88: For improved clarity it would be worthwhile to highlight that whilst specific quantitative guidance for SB is not included, the message is to reduce time spent in this behavior and replace with PA.**

We have included text in the Introduction to acknowledge the importance of reducing sedentary time and replacing with physical activity on page 3:

“Replacing sedentary time with physical activity may reduce the risk of cardiometabolic disease and mortality.^{2,9”}

- 5) Lines 90-9: Can the authors provide any data to support the ‘widespread use of commercial devices and smartphones’? For example, are there any data that can be included reported the number of devices used/bought/worn etc.**

We have included citation 11, which reported that over 1 in 4 Americans own a wearable device.

Corresponding text has been revised in the Introduction on page 3:

“The increasingly widespread use of commercial wearable devices (greater than 1 in 4 Americans) and smartphones that can track step count allows patients to monitor step count as a more accessible measure of physical activity.¹¹”

- 6) Fitbit data: Were only certain models/versions of Fitbit included in the data set or were different models permitted? If different models/versions are included, is there the possibility that algorithms for calculating activity metrics will have evolved over time? If this, this would need to be noted as a limitation**

Thank you for raising this concern. Participants were allowed to have different Fitbit devices. However, the Fitbit algorithms have not been changed since launching in 2017.

We have added this as a limitation in the Discussion on page 8:

“Additionally, participants were not restricted to a single model of Fitbit devices, but the Fitbit algorithm has not been changed since launching in 2017.”

- 7) Line 127-128: What was the justification for these criteria for a valid day? Can a reference/rationale be provided in the manuscript to support this?**

These were criteria for valid day that were drawn from prior studies with the Fitbit cohort in the *All of Us* research program. We have added relevant citations 15 and 19.

- 8) For the age categories, the cross over between categories makes is unclear how participants were grouped. For example, if someone was aged 40 years old, were they in the 18-40 or 40-60 category?**

The age categories excluded the right bound. We have updated the Table 1 to be clearer with the following age categories: 18-39, 40-59, 60-79, ≥ 80 .

- 9) For clarity, for alcohol classifications it would be beneficial to specify if the numerical values are referring to units or number of drinks.**

We agree that it would be helpful to have specific number of drinks. Unfortunately, the alcohol data was drawn from survey questions which had pre-defined classifications.

- 10) It is concluded that ‘results may help healthcare providers deliver more personalized, evidence-based recommendations tailored to individual behavior patterns and chronic diseases’. Given the variation in step count totals that were required to offset SB for different diseases, have the authors considered how practical this could be? Research has suggested that healthcare providers often do not know the global PA guidelines (i.e., 150mins per week), therefore, how feasible would it be for them to be recommending bespoke step counts for different SB profiles and health conditions? It would be beneficial to consider this further in the manuscript**

Thank you for raising the important question. In Figure 2, we see that risk reduction plateaued around 8,000 steps, which has been consistent with prior studies and further supports around 8,000 steps as a practical target for the general population. We also observe that those with greater sedentary time typically needed more daily steps to offset risk. Another approach would be to take the maximum number steps to offset risk of chronic disease for a given average sedentary time. For example, for patients with 14 hours of sedentary time, would need about 9,000 to 11,000 daily steps to offset risk of all the reported chronic diseases, which is consistent to the previously reported number of steps required to offset risk of mortality and cardiovascular disease in the UK Biobank (Ahmadi et. al, Reference 13).

Corresponding text can be found in the Discussion on page 7:
“For hypertension, heart failure, and MASLD, risk reduction plateaued around 8,000 steps. These findings support 8,000 steps as a practical target for health benefits, which is supported by recent studies that found between 7,000 and 9,000 daily steps is sufficient for most adults.^{13-16,34,35”}

11) Lines 252: Please amend typo ‘that that’

Thanks for catching. Typo was fixed.

12) Line 301: It would be beneficial to discuss this finding in relation to recent work suggesting 7,000 steps is sufficient for health outcomes (Daily steps and health outcomes in adults: a systematic review and dose-response meta-analysis, DOI:10.1016/S2468-2667(25)00164-1).

Thank you for the suggestion. We have revised the text in the Discussion as above in point 10 to discuss our suggested daily step count in the context of recent studies, including the suggested citation.

Thank you for taking the time to review our work.

Evan L. Brittain, MD, MSc
2525 West End Avenue, Suite 300A, Nashville, TN 37203
evan.brittain@vumc.org